# Relapse into Undernutrition in a Nutritional Program in HIV Care and the Impact of Food Insecurity: A Mixed-Methods Study in Tigray Region, Ethiopia

**DOI:** 10.3390/ijerph18020732

**Published:** 2021-01-16

**Authors:** Fisaha Haile Tesfay, Anna Ziersch, Sara Javanparast, Lillian Mwanri

**Affiliations:** 1Institute of Health Transformation, Deakin University, Burwood, VIC 3125, Australia; 2Southgate Institute for Health, Society and Equity, Flinders University, Bedford Park, SA 5042, Australia; anna.ziersch@flinders.edu.au; 3School of Public Health, Mekelle University, Mekelle, Tigray, Ethiopia; 4College of Medicine and Public Health, Flinders University, Bedford Park, SA 5042, Australia; sara.javanparast@flinders.edu.au (S.J.); lillian.mwanri@flinders.edu.au (L.M.)

**Keywords:** food insecurity, relapse, mixed methods, poverty, poor livelihood

## Abstract

The relapse into undernutrition after nutritional recovery among those enrolled in a nutritional program is a common challenge of nutritional programs in HIV care settings, but there is little evidence on the determinants of the relapse. Nutritional programs in HIV care settings in many countries are not well designed to sustain the gains obtained from enrolment in a nutritional program. This study examined relapse into undernutrition and associated factors among people living with HIV in the Tigray region of Ethiopia. The study employed a mixed-methods approach, involving quantitative and qualitative studies. Among those who graduated from the nutritional program, 18% of adults and 7% of children relapsed into undernutrition. The mean time to relapse for adults was 68.5 months (95% CI, 67.0–69.9). Various sociodemographic, clinical, and nutritional characteristics were associated with a relapse into undernutrition. A considerable proportion of adults and children relapsed after nutritional recovery. Food insecurity and poor socioeconomic status were a common experience among those enrolled in the nutritional program. Hence, nutritional programs should design strategies to sustain the nutritional gains of those enrolled in the nutritional programs and address the food insecurity which was reported as one of the contributors to relapse into undernutrition among the program participants.

## 1. Introduction

HIV and undernutrition are strongly interrelated, and act synergistically among people living with HIV [1]. Both undernutrition and HIV undermine the immune system, and significantly impact on disease progression, clinical outcomes, and the quality of life [2]. Undernutrition exacerbates the negative effect of HIV on the immune system by increasing patient’s vulnerability to AIDS and related opportunistic infections [1]. Similarly, HIV and AIDS contribute to weight loss (characterized by an involuntary weight loss of more than 10%) among people living with HIV [3,4]. Both asymptomatic and symptomatic HIV infection increases energy expenditure [5], contributing to weight loss. For example, the daily energy requirement of people living with HIV and AIDS is 30% higher than non-HIV infected people of the same age, sex, and level of physical activity [6].

The overall aim of nutritional programs in HIV care settings is to improve nutritional status, HIV treatment outcomes, and antiretroviral therapy (ART) adherence for people living with HIV. The types of nutritional programs and their implementation vary widely across countries. Many nutritional programs in sub-Saharan Africa (SSA) provide ready to use therapeutic foods (RUTF), or ready to use supplementary foods (RUSF), for a short period of time to tackle undernutrition amongst people living with HIV [1,2]. Alongside nutritional support, nutritional assessment and counseling are the essential components of nutritional programs in HIV care in SSA [3].

Most nutritional programs in SSA enroll people living with HIV who are undernourished, to treat undernutrition in order to achieve nutritional recovery. In Kenya [4] and Ethiopia [5], undernourished people living with HIV are enrolled in the nutritional program, and are provided with Plumpynut for at least six months for those who are severely undernourished, and Plumpysup for at least three months for those who are moderately undernourished. Although the nutritional programs seek to address acute undernutrition, there is a risk of relapsing once people exit the program. For instance, a study from Ethiopia found a high (20%) relapse into undernutrition after nutritional recovery [6]. Despite this, there have been no studies that directly examined the role of food security as a potential contributor to a relapse into undernutrition. However, food insecurity is a common occurrence among those enrolled in the nutritional program in HIV care in Ethiopia [6], and may be a key factor to consider.

Household food insecurity (referred to in this study as food insecurity) is generally defined as “the inability of the household to acquire the food needed by its members” [7]. Food insecurity is a broad concept with varying indicators across contexts, and usually depends on the target population and scale of measurements; for example, at country, household, and individual levels [8]. Broadly, food security encompasses the quantity and quality of food accessed by a country, household, or individual [7,8].

Evidence suggests a high prevalence of food insecurity among people living with HIV [9,10,11]. For example, Nagata and colleagues found that 79% and 21% of patients living with HIV in a study conducted in Kenya were severely and moderately food insecure, respectively [11]. Similarly, a study conducted in Ethiopia reported a high proportion (63%) of food insecurity among people living with HIV enrolled in HIV care [12].

Given the high magnitude of food insecurity, nutritional programs in SSA focus on addressing only the short term nutritional problems. For instance, research suggests that food insecurity is one of the major contributors to low attainment of nutritional recovery among people living with HIV enrolled in nutritional care [6,13]. Drop out (default) from a nutritional program in HIV care settings has also been found to be related to household food insecurity in SSA [6,13].

Nutritional programs in HIV care neither consider, nor assess individual or household food insecurity when enrolling people living with HIV into nutritional programs. For this reason, maintaining nutritional status after nutritional recovery is a major challenge for both adults and children enrolled in the programs [6,13,14,15]. However, there appears to have been no study that has examined the factors that are associated with relapse of undernutrition after nutritional recovery. Hence, this mixed methods study aims to address this gap by examining the extent and predictors of relapse into undernutrition for patients enrolled in a nutritional program in HIV care in Ethiopia.

## 2. Methods

### 2.1. Description of the Nutritional Program and Study Settings

This study was conducted in the Tigray region of Ethiopia−in three purposively selected general hospitals, namely, Mekelle, Lemlem Karl, and Shul. The study setting has previously been described elsewhere [5].

In Ethiopia, all HIV patients enrolled into chronic HIV care are regularly screened for undernutrition during their follow up visits to HIV care services. After enrolment in the nutritional program, sociodemographic, clinical, anthropometric, and nutritional outcome data are collected for monitoring and provision of nutritional support (RUTF or RUSF). RUTF is given to individuals with severe acute undernutrition, while RUSF is provided to moderate and mild undernutrition. The enrolment and exit criteria, and ration sizes are different for children and adult HIV patients enrolled in the nutritional program. Depending on the severity of undernutrition, adult and child patients are provided with two sachets of RUSF daily for a maximum of three months for moderate acute undernutrition (MAM), or body mass index (BMI) of ≤ 17.99. However, for individuals with severe acute undernutrition (SAM), or BMI ≤ 16, four sachets of RUTF are provided daily for a maximum of six months, together with nutritional counselling [6,16]. We have reported the nutritional outcomes after enrolment to the nutritional program elsewhere [5].

### 2.2. Study Design

This study employed a mixed-methods approach to data collection, using quantitative and qualitative data in a complementary way [17,18,19]. The mixing of the qualitative and quantitative studies was especially relevant to the interpretation stage. Further details of the study design of this paper have been reported elsewhere [5,20].

### 2.3. Sample Size and Recruitment

Quantitative and qualitative data were collected between March and July 2016. Quantitative data included records of 1756 adults and 236 children living with HIV (from November 2010 to February 2016). Data were retrieved from a paper-based database in the three selected hospitals. Records with no baseline demographic, socioeconomic, clinical, immunological, nutritional, and anthropometric characteristics were excluded; a total of 24 adult and 9 child records.

The qualitative study involved in-depth interviews with 48 purposively selected individuals; 20 adults living with HIV, 15 caregivers of children living with HIV, 11 health providers, and 2 program managers. Adults and caregivers were recruited for the in-depth interviews at the health facility during their regular follow up for HIV services, including the nutritional program. Health providers provided information to them about the project, and invited them to contact the researcher to participate in the research if interested. They were assured that their participation or otherwise in the research would be confidential, and have no impact on their access to services. Health providers and program managers working in the program for more than 1 year were recruited through an invitation letter, with contact details of the researcher posted on a notice board in the health facility. Interested health provider participants contacted the researcher. Participant recruitment continued until data saturation was achieved. Interviews were conducted at a venue convenient to all participants. Pregnant women were not included in the current study because: (a) the sample size was small, (b) separate anthropometric measures are used for pregnant mothers.

### 2.4. Data Collection Procedure

Quantitative data were retrieved from multiple sources enabling cross-matching and included: (a) nutritional program follow up form; (b) HIV care follow up form, and (c) ART pharmacy database. Upon an HIV diagnosis and enrolment to HIV care (ART and Pre-ART services), each patient is assigned a unique identification number that links them with the HIV care service.

The patient’s unique identification number was used to match or relate each individual patient’s data from the various sources. The qualitative data were collected using interview guides covering issues around participants’ knowledge of the nutritional program, and factors enabling or constraining program utilization. In-depth interviews were conducted by the first author (FT) in the local language, Tigrigna, and audio-recorded. Comprehensive field notes were also taken during the fieldwork. Further details on data collection procedures for both the quantitative and qualitative studies have been published elsewhere [5,20].

### 2.5. Data Management and Analysis

#### 2.5.1. Quantitative Data

Quantitative data analysis involved a description of the demographic, socioeconomic, clinical, immunological, nutritional, and anthropometric characteristics of adult and child records. In addition, a description of the outcome variables, such as relapses of undernutrition and frequency of relapse into undernutrition was made.

We conducted bivariate and multivariate Cox regression to identify the independent predictors of relapses into undernutrition after nutritional recovery. The bivariate Cox regression analysis was used to estimate the crude hazard ratio. Independent variables that were statistically significant at *p* < 0.3 were taken to multivariate Cox regression to estimate the adjusted hazard ratio, with 95% CI. In multivariate Cox regression, statistical significance was declared at *p* < 0.05, and the hazard ratio was used to interpret the results.

The second outcome variable was the frequency of a relapse into undernutrition. To identify the determinants of frequency of a relapse into undernutrition, bivariate logistic regression was conducted to estimate the crude odds ratio, and statistical significance was considered at *p*-value < 0.3. Assumptions of logistic regression, such as multicollinearity and homogeneity of variance, were checked, and no collinearity was identified. Factors that were found to be statistically significant at *p* < 0.3 were taken to the multivariate logistic regression model to determine the independent predictors of the frequency of a relapse. Adjusted odds ratio with 95% CI were used to determine the magnitude and direction of the relationship, and statistical significance was considered at *p* < 0.05.

#### 2.5.2. Qualitative Data

Qualitative interviews were translated and transcribed into English by the first author (FT). Framework thematic analysis [21] was used to analyze the qualitative data. A coding framework was developed and discussed in the team meetings until a consensus was reached among the authors. Three interviews were double coded by the authors, and differences were discussed until consensus was reached. Data were analyzed using QSR NVivo, and themes and categories emerged after thorough reading and understanding of the interview data, field notes, and memos. Illustrative quotes are used to describe themes and categories.

### 2.6. Operational Definition

Relapse into undernutrition: participants returned to the nutritional program with either SAM or MAM after they had been declared nutritionally recovered. In the case of children living with HIV, a child returned to the clinic with either SAM or MAM after they had been declared recovered, and had been discharged from the nutritional program. 

## 3. Results

### 3.1. Results of the Quantitative Study

#### 3.1.1. Demographic and Socioeconomic Characteristics of Adults and Children Living with HIV

The most common age group for adults was the 26–35 age group (42.6%), 63% were female and two-thirds lived in urban areas. More than 40% were married, and over three-quarters (76.6%) had children, with the majority of people living in a household of less than five people. Nearly 62% of adults had attended at least primary and secondary education, and a large proportion of adults and parents of children living with HIV were unemployed. The vast majority of people reported their religion as Orthodox, or other Christian, religion. Over three quarters had disclosed their positive HIV status to at least someone, and 81% of adults were not members of a community HIV support group (Table 1). Demographic characteristics of adults living with HIV has also been published elsewhere [5].

The most common age group for children was 5–10 years (43.2%), and 47.5% were female. The vast majority (82.6%) of children lived with their parents (of whom more than half were living together), and three-quarters lived in urban areas. There were similar proportions of children who were the first, second, third, and fourth-child in their family. Most children attended school (Table 1).

#### 3.1.2. Clinical and Immunological Characteristics of Adults and Children Living with HIV

The functional status of the majority of adult participants was “working” (83.8%), while 50.5% of adults and 56% of children were in WHO clinical stage I. Furthermore, 94% of adults and 88% of children were on ART at baseline or at the time of enrolment in the nutritional program. In addition, 45.7% of adults were anemic at enrolment. While, 64% of adults and 72% of children had been on ART for 24 months at enrolment to the nutritional program. A quarter (25%) of adults and 22% of children had opportunistic infections, with TB being the major opportunistic infection in both groups (Table 2). The clinical and immunological characteristics of adults living with HIV have been published elsewhere [5].

#### 3.1.3. Nutritional and Anthropometric Characteristics of Adult and Child Participants

Before enrolment into the nutritional program, 62.5% of adults and 80% of children participants had MAM, while 23% of adults and 8% of children had SAM. Almost 78% of adults received less than three sachets per day during their engagement with the program. Having a good appetite was one of the criteria for enrolment into the nutritional program, and almost all of the adults reported having a good appetite. Those with poor appetite were treated for its causative agent (usually opportunistic infections) before enrolment into the nutritional program. The nutritional and anthropometric characteristics of adults have also been published elsewhere [5].

After enrolment in the nutritional program, 55% of adults and 71% of children recovered or achieved the graduation criteria of the targeted outcome (BMI of ≥ 18 kg/cm^2)^. Non-response (not meeting the target BMI) was identified in 21% of adults and 14% of children. Amongst the study cohort, 18% of adults and 14% of children failed to complete the program. Among those who achieved nutritional recovery, 170 (18%) adults and 12 (7%) children later relapsed (Table 3).

#### 3.1.4. Determinants of Relapses into Undernutrition

The total time to relapse after nutritional recovery in the nutritional program was 68.5 months (95% CI 67.0–69.9).

Educational status, membership of an HIV community support group, duration on ART, ambulatory and bedridden functional status, presence of opportunistic infections, and moderate and severe acute undernutrition were significantly associated with relapse into undernutrition after nutritional recovery. Adults who attended primary and secondary education were 2.8 and 3.7 times more likely to relapse than those who attended tertiary and above, but there was no difference in those who had no education. Non-membership of HIV community support group was associated with a 1.7 times higher chance of relapse than those who were members of a community HIV support group. In addition, those who had been on ART for longer periods were more likely to relapse than those who had been on ART for less than 24 months. Adults who had opportunistic infections at baseline were at 1.7 times the risk of relapse into undernutrition after nutritional recovery (Table 4).

#### 3.1.5. Frequency of Relapses of Undernutrition

Data related to the frequency of relapses into undernutrition were limited, as they were only available from September 2012 to July 2016. Of the 170 adults who relapsed after recovery, 124 (72.9%) relapsed once, and 31 (18.3%) relapsed twice. The remaining, 10 (5.9%) and five (2.9%) relapsed three and four times, respectively, after nutritional recovery. There was not a large enough sample for children to run a multivariate analysis.

In the multivariate logistic regression, where the statistical significance was declared at *p* < 0.05, place of residence, education, employment status, functional status, presence of anemia, and baseline nutritional status were statistically significantly associated with frequency of relapse into undernutrition. Rural dwellers were three times more likely to relapse more than once when compared to those from urban areas. Those who had not attended formal education were more likely to relapse more than once, compared to those who attended secondary education. There was no difference between those attending primary, and secondary education and above. Regarding employment, those who were employed were more likely to relapse more than once than those who were unemployed. Individuals who were bedridden were five times more likely to relapse more than once than those who were healthy. Individuals who were anemic at enrolment were nine times more likely to relapse more than once than those who were not. Those who were severely and moderately undernourished at baseline were four and nine times, respectively, more likely to relapse more than once than those who had mild undernutrition (Table 5).

### 3.2. Results of the Qualitative Findings

In the qualitative analysis, food insecurity emerged as one of the key factors that directly and/or indirectly influenced program outcomes through: initial undernutrition and motivation to enroll in the program, nutrition program use in terms of selling and sharing the nutritional support, a disincentive to graduate from the program, as well as eventual relapse.

#### 3.2.1. Food Insecurity, Poverty, and Undernutrition

As demonstrated by the quote below, adults cited household food insecurity as a key cause of weight loss.

*“Yes, I made an effort to maintain my weight, but my economic status determines my nutritional status. If I didn’t have enough, then from where can I get it? I try (to get) by what I have but my economic status is very low. I know if I eat properly, I will have the energy to work and perform like others but if I am poor, I can do nothing”* (Adult female, age 48 #9)

Poverty and HIV seemed to be interwoven, both contributing to food insecurity and undernutrition among people living with HIV. Many adults, and child participants’ caregivers, reported the absence of adequate educational attainment, lack of regular and reliable employment, and a lack of sustainable and reliable income to support them and their family.

*“If you are poor and don’t have a good job, even if you try to create your own job, no one allows you to work for the reason of no education. I asked the kebele administration for a job but they told me that I have no education. This makes me very angry. Otherwise, if I have a job, I will not lack adequate and balanced food.”* (Adult female, age 31 #1)

This lack of access to adequate food was one of the key motivations for adults and caregivers to enroll in the nutritional program. In particular, those who lived in urban areas and did not have a reliable source of income or were unemployed reported the lack of adequate food in their household as a key reason for their enrolment in the nutritional program. These motivations were shared by both male and female adult participants, reporting circumstances of food shortage in the household.

*“If you have something to eat, it should be fine but if you don’t have anything to eat like me this food support is very important.”* (Adult male, age 40 #18)

*“Yes… many people suffer from a lack of access to adequate food. Nutritional support is very helpful to all HIV patients. I have benefited a lot from nutritional support. So, I suggest this nutritional support to all people with HIV.”* (Adult female, age45 #2)

#### 3.2.2. Food Insecurity as a Driver of Selling and Sharing of the Nutritional Support

Poor household economic situation was one of the main reasons for people to sell or share nutritional support. A health provider and program manager stated that people sell nutritional support because of their economic conditions, and to exchange for other household consumables such as oil or coffee beans:

*“Instead of eating it (nutritional support) for themselves only, they want to sell and exchange it for other household needs such as sugar, salt or oil. Most of the time, this is the reason but the base is the poor livelihood condition. So, they are not doing it intentionally, but it is because of their problems.”* (Health provider #9)

*“Even though the patients were counseled well, one reason for selling could be the existing economic problems. Poverty by itself would encourage individuals to sell it (the nutritional support) and spend the money on something that matters to the family is there.”* (Program manager #1)

Health providers, as well as adults and caregivers, also reported food insecurity as a leading reason for sharing practices. The following quotes below illustrate these assertions.

*“Yes, there is sharing among household members. As far as there is an economic problem, it is not necessarily selling but also there is sharing. Because if it is given to him, it is likely that the mother will share it with his siblings. If given two sachets then the mother gives one to her other child and keeps one for the HIV positive child.”* (Health provider #7)

*“Even though it is prescribed to the sick child if there is no adequate food to eat in the household the mother may share it to other children. So, the mother shares it to fulfill the dietary needs of the other children in the household because she has nothing to give to the other child.”* (Caregiver, age 35 #9)

*“Even now, I can’t get enough (food) and I am not taking adequate food. I give everything (including RTUF) I have to my children and my main effort is to feed and care for them. With all the problems I have, I can’t get enough food to support myself and my children”* (Adult female, age 29 #15)

#### 3.2.3. Food Insecurity, Dependence, and Disincentive to Graduate from the Program

Some health providers expressed concerns about potential dependence on the nutritional program that people enrolled could develop due to poverty and food insecurity.

*“Some patients become very reliant on the supports (soaps, water treatment jerrycan, and the nutritional support) given from the health facility because they believe they should get supported due to their HIV condition.”* (Health provider #2)

A reluctance to graduate from the program among some participants enrolled in the nutritional program due to poor socioeconomic status was also voiced as a concern by health providers, who reported that participants preferred prolonged enrolment in the nutritional program to compensate for poor household socioeconomic status:

*“It is because of their poor economic status that most don’t want to graduate. If he graduates, I will not give him the Plumpynut next time because they don’t have other sources of income. Most want their weight to stay as low as possible.”* (Health provider, age 35 #7)

#### 3.2.4. Food Insecurity as a Contributor to Relapse

Despite the high magnitude of food insecurity among those enrolled in the nutritional program, health providers identified no remedial strategies to prevent relapses of undernutrition among those who graduated as part of the program.

*“The major issue that creates a problem in this regard is that when they (people living with HIV) graduate from the nutritional program. There is no local or international NGO with which we can link people after their graduation from the program. There are no more NGOs in our area and I don’t know the reason. So, there are no efforts made to prevent relapse into malnutrition after nutritional recovery. Because they should be linked to other income-generating activities.”* (Health provider, age 27 #2)

This was highlighted as a key improvement needed with the program:

*“Things that need improvement in the nutrition program, now after you treat him for malnutrition and he graduates, there is nobody who helps you prevention of relapse. At least you have to link the patients in order for the problem not to come again. So, we have a big problem in this regard and we don’t have a supportive organization to do this.” (Health provider, age 37 #6)”* (Health provider #2)

As part of improving food insecurity, key suggestions were the provision and maintenance of land in rural areas to support participants of the program, as well as that initiatives prioritizing adults living with HIV in income-generating activities were unavailable for patients living in urban areas.

*“I heard there is some support for people living in a rural areas. So they should do a similar way in urban areas. Living in a city with HIV is very difficult and creates a problem of house rent and others.”* (Adult female, age 48 #9)

## 4. Discussion

This study highlights that a relapse after nutritional recovery is an important issue for adults and children enrolled in nutritional programs. The quantitative analysis found a range of demographic, socioeconomic, clinical, and nutritional characteristics contributed to the relapse into undernutrition among adults living with HIV. The qualitative component also highlighted the potentially important role of food security.

It was obvious, especially as detailed in the qualitative findings, that food insecurity was an important motivator for participants to enroll in the nutritional program. Similarly to the findings reported in previous studies, factors such as low income and unemployment contributed to food insecurity among adults and caregivers living with HIV. For instance, studies in sub-Saharan Africa [22,23,24] indicated that people living with HIV are more likely to be unemployed and less skilled. Consistent with earlier studies, the current study demonstrated high rates of unemployment, as supported by the quantitative findings and narrated by many participants in the qualitative interviews. No income or unreliable sources of income could be related to the lack of employment, which seemed to place significant stress on household income, contributing to the vicious cycle of food insecurity.

While both adults and children in the study regained weight and achieved nutritional recovery after enrolment in the nutritional program, sustaining nutritional wellbeing in the presence of household food insecurity was a challenge. In the current study, 18% of adults and 7% of children relapsed after nutritional recovery, and 27% of adults relapsed more than once. This vicious cycle of relapses can be detrimental, not only to the program, but also to the affected individuals, as long lasting undernutrition leads to poor physical strength (leading to poor contribution to economic activities) and can be life threatening, especially in children [25]. In a further multivariate analysis, factors such as demographic, socioeconomic, clinical, and nutritional factors contributed to the relapse into undernutrition after nutritional recovery.

The quantitative analysis found that those with lower levels of education were more likely to relapse than those who attended a tertiary level of education. Similarly, when considering the frequency of relapse, those who did not attend formal education were more likely to relapse more frequently than those who attended tertiary education. This finding is novel. We could not find previous studies that have examined the relationship between educational status and relapse into undernutrition, although it is well acknowledged that, in general, better education is related to better health [26,27,28], and greater education could be related to enhanced understanding of the program, including the nutritional counseling, and strategies to prevent relapses. Better education may also create a better opportunity for employment, and this may contribute to better access and availability of food.

The study findings also indicated that those who were not employed were more likely to relapse, supporting a reflection of the impact of lower income on food security, as households would need income to improve their purchasing power. Surprisingly, those who were employed were more likely to relapse more than once compared to those who were not employed. It is uncertain why this is, further research is needed to explore this potential relationship. From the qualitative findings of the current study, precarious employment, such as subsistence farming and daily laborers, were reported to cause difficulties in keeping up with schedules and medical appointments, which could possibly explain this scenario.

The findings from the current study also highlighted that rural dwellers were more likely to relapse multiple times than their urban counterparts. Another study from Ethiopia and Uganda also found that people from rural areas have a more limited access to basic services such as health, education, and other social services than their urban counterparts [29], and hence lack of basic social services, such as access to transportation to the health facility, may contribute to relapse among those living in rural areas.

Another interesting finding of this study was that adults involved in community HIV support groups were less likely to relapse. Involvement in community HIV support groups may create an opportunity to discuss and receive counseling on how to manage nutrition, offering a greater support for participants to maintain weight [30,31].

In addition, clinical and nutritional factors such as the duration of ART, presence of opportunistic infections, and anemia were associated with relapse into undernutrition. For example, anemia among people living with HIV is common due to the HIV illness and some ART medications [32,33]. Similar findings were also demonstrated in the current study, with those presenting with anemia and severe and moderate acute undernutrition at enrolment being more likely to relapse. Furthermore, those with the worst clinical conditions (bedridden functional status) were more likely to relapse more than once compared to those who were categorized as apparently healthy, because a progressive course of HIV infection contributes to undernutrition

These clinical and nutritional conditions may directly contribute to undernutrition, because the worst HIV infections contribute to undernutrition [34]. It may also reflect food insecurity, where such conditions may limit the individual’s economic contribution, leading to poverty and food insecurity. This is consistent with the literature, where prolonged illnesses have been found to be essential contributors to poverty and food insecurity among people living with HIV [35,36]. There are no other studies that have examined the role of these clinical and nutritional characteristics in contributing to relapse into undernutrition after nutritional recovery among those enrolled in the nutritional programs.

Additional findings revealed that sharing the nutritional support was a common practice, and appeared to occur in instances of household food insecurity. In the current study, adults and caregivers shared the nutritional support to fulfill the dietary needs of their families due to a lack of adequate food in the household. These findings support a study conducted in Ethiopia that reported RUTF sharing was more common in poor households, indicating the negative impact of food insecurity on the nutritional program [37]. Although no program participants reported selling the nutritional support, health providers stated that this happened, and was also done for people to fulfill family nutritional needs.

While undernutrition in people living with HIV is underpinned by underlying food insecurity and the determinants of food insecurity, it seems that nutritional programs primarily focus on short term treatment of undernutrition, dealing with the “tip of the iceberg”. As a result, despite adults and children living with HIV recovering from undernutrition after enrolment in the nutritional program, relapsing is likely to be a common scenario if food insecurity is not addressed.

While the study has provided important qualitative and quantitative evidence regarding the predictors of relapse and the role of food insecurity in contributing to relapse into undernutrition, there were a number of limitations. While the relationship between food insecurity and relapse emerged during the qualitative analysis, it was not directly measured in the quantitative component. Additionally, it was not possible to validate the accuracy and precision of measurement of the demographic, socioeconomic, clinical, and nutritional characteristics, as only secondary data was used. It is also likely that the originality of information in the qualitative study may have been lost during the translation and transcription, even though efforts were made to maintain the original conversation, as narrated by the participants. It was also not possible to calculate the average time to relapse in children because of the small sample size

## 5. Conclusions

In conclusion, a significant proportion of adults and children relapsed after nutritional recovery, and qualitative findings highlighted that food insecurity was common among people living with HIV enrolled in the nutritional program. Various sociodemographic and cultural characteristics, such as low education, poor employment, rural residence, and involvement in a community support group, were predictors of relapse. Moreover, clinical and nutritional characteristics, such as poor clinical condition and presence of opportunistic infections and SAM/MAM, were associated with relapse into undernutrition. The above factors likely contributed to a vicious cycle of complex poor nutritional outcomes. Thus, addressing poverty and poor livelihood is crucial to improving and sustaining the nutritional wellbeing of people living with HIV, and in reducing the chances of relapsing into undernutrition post nutritional recovery. To improve the overall nutritional wellbeing and HIV treatment outcomes in these programs, there is a need to consider strategies that sustain the nutritional gains during their enrolment in the nutritional program. Further research should be conducted on key strategies to sustain nutritional gains after nutritional recovery, and on how to address undernutrition.

## Figures and Tables

**Table 1 ijerph-18-00732-t001:** Demographic and socioeconomic data of adults and children enrolled in the nutritional program.

Adults (*n* = 1757) ^1^				Children (*n* = 236)			
Variables	Categories	Number	Percent	Variables	Categories	Percent	Number
Age	<25	216	12.3	Age	<5 years	62	26.3
	26–35	749	42.6		5–10 years	102	43.2
	36–45	531	30.2		>10 years	72	30.5
	>46	261	14.9	Sex	Male	124	52.5
Sex	Male	649	36.9		Female	112	47.5
	Female	1108	63.1	Residence	Urban	179	75.8
Residence	Urban	1171	66.6		Rural	57	24.2
	Rural	586	33.4	Child lives with	Parents	195	82.6
Marital status	Never married	265	15.1		Guardian	15	6.4
	Married	722	41.1		Grand parents	2	0.8
	Separated	164	9.3		In orphanage	14	5.9
	Divorced	381	21.7		Siblings	10	4.2
	Widow/Widower	225	12.8	Marital status of parents, *n* = 195	Mother and father live together	100	51.3
Education status	No education	535	30.4		Divorced	16	8.2
	Primary	637	36.3		Widowed	5	2.7
	Secondary	459	26.1		Single parent father	18	9.2
	Tertiary	126	7.2		Single parent mother	54	27.7
Religion	Orthodox and other Christian	1659	94.4	Mother alive, *n* = 208	Yes	175	84.1
	Muslim	98	5.6		No	33	15.9
Employment	Working	871	50.8	Emploment status of mother if alive. *n* = 147	Employed	55	37.4
	Employed but not working due to ill health	111	6.5		Unemployed	92	62.6
	Unemployed	732	42.7	Father alive, *n* = 202	Yes	146	72.3
Household family size	≤5	1543	87.8		No	56	27.7
	>5	207	11.8	Employment status of father if alive. *n* = 126	Employed	90	71.4
Have children	Yes	1346	76.6		Unemployed	36	28.6
	No	411	23.4	Child’s birth order	First	32	13.6
Membership of HIV related community support group	Yes	333	19.0		Second	42	17.8
	No	1424	81.0		Third	32	13.6
Disclosure to at least someone	Yes	1366	77.7		Fourth and above	28	11.9
	No	391	22.3	Child attend school	Yes	170	72.0
Name of hospital	Mekelle	1045	59.5		No	66	28.0
	Lemlem Karl	378	21.5	Reason for not attending school, *n* = 66	Too young	60	90.1
					Lack of fund	6	9.1
	Shul Hospital	334	19.0	Name of hospital	Mekelle	189	80.1
					Lemlem Karl	24	10.2
					Shul	23	9.7

^1^ The table on demographic and socioeconomic characteristics of adults are published in another related paper [5].

**Table 2 ijerph-18-00732-t002:** Clinical and immunological characteristics of adults and children enrolled in the nutritional program.

Adults (*n* = 1757) ^1^				Children (*n* = 236)			
Variables		Number	Percent	Variables			
Functional status	Working	1473	83.8	WHO clinical stage	Stage I	132	55.9
	Ambulatory	207	11.8		Stage II	42	17.8
	Bedridden *	77	4.4		Stage III	50	21.2
WHO clinical stage	Stage I	888	50.5		Stage IV	12	5.1
	Stage II	225	12.8	ART status at enrolment	On pre-ART	29	12.3
	Stage III	518	29.5		On ART	207	87.7
	Stage IV *	126	7.2	Duration on ART	<12 months	27	13.0
Baseline CD4 count	<200	675	39.0		13–24 months	31	15.0
	200–349	488	28.2		>24 months	149	72.0
	350–500	294	17.0	Contrimoxzole propylaxis	Yes	209	88.6
	>500	274	15.8		No	27	11.1
Presence of anemia	Anemic	687	45.7	Presence of opportunistic infection	Yes	52	22.0
	Not Anemic	815	54.3		No	184	78.0
ART status	Pre-ART	100	5.7	Type of opportunistic infection	TB	14	26.9
	ART	1657	94.3		TB and other	10	19.2
Duration on ART	<6 months	324	19.6		Other	28	53.8
	6–12 months	99	6.0	Eligibility criteria to ART	WHO clinical stage only	28	11.9
	12–24 months	175	10.6		CD4 or TLC only	15	58.5
	>24 months	1059	63.9		CD4% only	47	19.9
Presence of opportunistic infection	Yes	436	24.8				
	No	1321	75.2				
Type of opportunistic infection	TB	152	37.3				
	TB and others	47	11.5				
	Others	208	51.1				

Stage IV and bedridden indicate severe clinical condition. ^1^ The table on clinical and immunological characteristics of adults are published in another related paper [5]. * Bedridden refers to the severity of HIV sickness of people living with HIV. Bedridden is defined as spending 50% of the a month on bed due to HIV related sickness among people living with HIV.

**Table 3 ijerph-18-00732-t003:** Nutritional characteristics of adults and children enrolled in the nutritional program.

Adults (*n* = 1757)				Children (*n* = 236)			
Variables		Number	Percent	Variables		Number	Percent
Nutritional status at enrolment	Mild acute undernutrition	253	14.4	Nutritional status at enrolment	Mild acute undernutrition	29	12.3
	Moderate acute undernutrition	1098	62.5		Moderate acute undernutrition	188	79.7
	Severe acute undernutrition	406	23.1		Severe acute undernutrition	19	8.1
Number of sachets/day	≤3	1362	77.6	Duration on nutritional program	<3 months	28	11.9
	≥4	394	22.4		3 months	193	81.8
Appetite test done	Yes	1731	98.5		>3 months	15	6.4
	No	26	1.5	Nutritional outcome	Graduated	167	70.8
Appetite test results	Good	1150	65.5		Non-respondent	33	14.0
	Poor	580	33.0		Defaulted	33	13.9
Subsequent weight gain	Good	1650	93.9		Died	2	0.8
	Poor	70	4.0		Transferred out	1	0.4
Nutritional outcome	Graduated/recovered	971	55.3	Relapse into malnutrition	Yes	12	7.2
	Non-respondent	379	21.0		No	154	92.8
	In completed (Defaulted)	329	18.7				
	Death	35	2.0				
	Transferred out	43	2.4				
Relapse after nutritional recovery (*n* = 968)	Yes	170	17.6				
	No	801	82.4				

**Table 4 ijerph-18-00732-t004:** Determinants of relapse into undernutrition after nutritional recovery in adults.

Variables		Relapse into Malnutrition		Crude Hazard Ratio (95% CI)	Adjusted Hazard Ratio (95% CI)	*p*-Value
		No (%)	Yes (%)			
Sex	Male	289 (83.5)	57 (16.5)	0.92 (0.67–1.27)	0.91 (0.64–1.29)	0.591
	Female	509 (81.8)	113 (18.2)	1.0	1.0	
Residence	Urban	533 (81.5)	121 (18.5)	1.16 (0.83–1.62)	0.94 (0.65–1.34)	0.740
	Rural	265 (84.4)	49 (15.6)	1.0	1.0	
Marital status	Single	131 (86.8)	21 (13.2)	1.0	1.0	
	Married	342 (83.2)	69 (16.8)	1.33 (0.81–2.19)	1.27 (0.76–2.13)	0.363
	Divorced	228 (80.9)	54 (19.1)	1.49 (0.89–2.49)	1.50 (0.88–2.56)	0.132
	Widowed	97 (78.2)	27 (21.8)	1.57 (0.88–2.82)	1.36 (0.74–3.43)	0.331
Educational status	No education	231 (84.9)	41 (15.1)	1.055 (0.75–1.49)	2.84 (0.87–9.26)	0.080
	Primary	276 (79.8)	70 (20.2)	1.48 (1.07–2.03)	3.68 (1.15–11.77)	0.033
	Secondary	276 (79.8)	70 (20.2)	1.39 (1.0–1.93)	3.25 (1.01–10.48)	0.049
	Tertiary	291 (83.1)	59 (16.9)	1.0	1.0	
Employment	Working	425 (84.7)	77 (15.30	1.0	1.00	
	Not working	353 (79.3)	92 (20.7)	1.36 (1.00–1.84)	1.32 (0.97–1.81)	0.08
Membership of community support	Yes	127 (73.0)	47 (27.0)	1.0	1.0	
	No	671 (84.50)	123 (15.5)	1.87 (1.33–2.63)	1.78 (1.25–2.54)	0.001
Functional status	Working	694 (81.9)	153 (18.1)	1.0	1.0	
	Ambulatory and bedridden	104 (86.0)	17 (14.0)	0.76 (0.46–1.26)	5.2 (1.63–16.67)	0.005
Duration on ART	≤6 months	133 (88.1)	18 (11.9)	1.0	1.0	
	6–12 months	52 (91.2)	5 (8.8)	0.70 (0.26–1.89)	0.83 (0.3–2.25)	0.71
	13–24 months	80 (87.0)	12 (13.0)	1.0 (0.48–2.07)	1.28 (0.61–2.70)	0.52
	>24 months	495 (79.6)	127 (20.4)	1.78 (1.08–2.91)	2.15 (1.27–3.63)	0.004
Opportunistic infection	Yes	182 (77.8)	52 (22.2)	1.38 (1.0–1.91)	1.68 (1.18–2.39)	0.004
	No	616 (83.9)	118 (16.1)	1.0	1.0	
Baseline nutritional status	Mild	186 (80.9)	44 (19.1)	1.0	1.0	
	Moderate	543 (83.3)	109 (16.7)	0.85 (0.60–1.21)	0.93 (0.65–1.33)	0.008
	Severe	68 (80.2)	17 (17.8)	1.09 (0.62–1.90)	1.34 (0.62–2.08)	0.003

**Table 5 ijerph-18-00732-t005:** Determinants of frequency of relapse into undernutrition after nutritional recovery.

Variables		Relapsed Only Once (%)	Relapsed More Than Once (%)	Crude Odds Ratio (COR) (95% CI)	Adjusted Odds Ratio (AOR) (95% CI)	*p*-Value
Place of residence	Urban	95 (78.5)	26 (21.5)	1.0	1.0	
	Rural	29 (59.2)	20 (40.8)	2.52 (1.23–5.16)	3.14 (1.30–7.57)	0.011
Marital status	Single	17 (85.0)	3 (15.0)	0.6 (0.13–2.72)	0.49 (0.076–3.19)	0.463
	Married	46 (65.7)	24 (34.3)	1.74 (0.62–4.91)	1.50 (0.40–5.62)	0.551
	Divorced	41 (75.9)	13 (24.1)	1.06 (0.35–3.19)	0.6 (0.15–2.35)	0.464
	Widowed	20 (76.9)	6 (23.1)	1.0	1.0	
Educational status	No education	25 (61.0)	16 (39.0)	5.65 (1.98–16.19)	3.88 (1.22–12.36)	0.022
	Primary	46 (65.7)	24 (34.3)	4.61 (1.73–12.25)	2.23 (0.75–6.58)	0.150
	Secondary and above	53 (89.8)	6 (10.2)	1.0	1.0	
Employment status	Working	42 (53.8)	36 (46.2)	7.81 (3.44–17.73)	3.86 (1.15–12.94)	0.029
	Not working	82 (90.1)	9 (9.9)	1.0	1.0	
Disclosure of HIV status	Yes	94 (69.6)	41 (30.40	2.62 (0.95–7.22)	2.44 (0.79–7.53)	0.121
	No	30 (85.7)	5 (14.3)	1.0	1.0	
Functional status	Working	117 (76.5)	36 (23.0)	1.0	1.0	
	Ambulatory or bedridden	7 (41.2)	10 (58.8)	4.64 (1.65–13.08)	5.21 (1.63–16.67)	0.005
WHO clinical stage	Stage I and II	89 (78.1)	25 (21.9)	1.0	1.0	
	Stage III and V	35 (62.5)	21 (37.5)	2.14 (1.06–4.30)	1.26 (0.45–3.55)	0.673
Duration on ART	≤6 months	9 (50.0)	9 (50.00)	3.38 (1.24–9.17)	2.51 (0.79–8.00)	0.121
	>6 months	111 (77.1)	33 (22.9)	1.0	1.0	
Haemoglobin level	Anemic	30 (51.7)	28 (48.3)	5.54 (2.61–11.74)	9.28 (3.39–25.33)	0.0001
	Not anemic	89 (85.6)	15 (14.4)	1.0	1.0	
Cotrimoxazole	Yes	81 (69.2)	36 (30.8)	1.91 (0.86–4.22)	1.86 (0.69–5.04))	0.223
	No	43 (81.1)	10 (18.9)	1.0	1.0	
Baseline nutritional status	Mild malnutrition	39 (88.6)	5 (11.4)	1.0	1.0	
	Moderate malnutrition	75(68.8)	34 (31.2)	3.54 (1.28–9.76)	4.30 (1.37–13.54))	0.013
	Severe malnutrition	10(58.8)	7 (41.2)	5.46 (1.43–20.88)	9.90 (2.04–48.14)	0.004

## Data Availability

The data presented in this study are available on request from the corresponding author. The data are not publicly available due to for ethical reasons. Because I promised the Ethical review boards of Social and Behavioural Research Ethics of Flinders University and Ethical Review Committee of Mekelle University not share the data without the consent of participants for a third party.

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
