# Peer review of "Relapse into Undernutrition in a Nutritional Program in HIV Care and the Impact of Food Insecurity: A Mixed-Methods Study in Tigray Region, Ethiopia"

_ijerph, 2021, doi:10.3390/ijerph18020732_

Round 1

Reviewer 1 Report

The authors have carried out qualitative and quantitative analysis of HIV positive individuals enrolled in HIV care in order to assess factors involved in the relapse to undernutrition after achieving nutritional recovery in Ethiopia. Undernutrition and malnutrition are major complications of – and significant factors that contribute to the rapid progression of AIDS, and is indeed very common in Africa. Identifying factors involved in the relapse would indeed be beneficial, aiding in the optimization of nutritional programs focusing on HIV care. This study appears to be an extended analysis of an already published article by the authors, using the same data from the adult population; however, the statistical analysis has been modified to fit the purpose of the study.

The manuscript is very well written and easy to digest, although some spelling mistakes and grammatical errors are present and need addressing; such as:

  • Relapse to or into, instead of relapse for throughout the manuscript.

Line 48: „ However, other factors that were 49 associated with relapse of undernutrition were not examined”

50: Household food insecurity (referred to in this study as food insecurity) is generally defined as  “the inability  of the household to acquire the food needed by its members”

79: ” their follow up visits to HIV care services”

93: “especially relevant to the interpretation stage”

130: “of relapses to undernutrition” and in other sentences as well

Table 1. “Demographic and socioeconomic data of adults and children enrolled in the nutritional program”

Table 2. Clinical and immunological characteristics of adults and children enrolled in the nutritional program.

“Those with poor appetite were treated for the causative agent (usually opportunistic infections)”

206: “more likely to relapse than those who attended tertiary and above, but there was no difference in  those who had no education”

Table 4: “YES %”

218: “times respectively after nutritional recovery. There was not enough sample for children to run”

290: enrolled

239: diseninctive?

253: balanced

334: “could be related to the lack of employment”

346: when considering the

377: “because a progressive course of HIV infection contributes to undernutrition

385 “appeared to occur in households with food insecurity

390: done by people

Questions and comments:

357: income to improve the purchasing power. Surprisingly, those who were employed were more likely to relapse more than once compared to those who were not employed:

  • Isn’t it plausible that the type and nature of the employment can affect the likelihood to relapse? Was there any analysis as to whether physical/rural/agricultural work contributed to the relapse to undernutrition?

  • Also, was the pregnancy of female participants controlled for during the study?

116: “Unique ART and pre-ART number were used to match or relate data from the various sources”:

  • What does this mean?

  • The quotes in the qualitative analysis are understandable, and I get that they were literal translations, however, I think that the statements should be worded in a better form, not necessarily meaning that they should be reworded, but at least in a grammatically correct form.

  • Table 1, and 2 are identical to the ones previously published by the authors when studying factors influencing the effectiveness of nutritional programs (Reference 5), with the exception that they also include data from the pediatric population. I understand that the population is the same (at least for the adult population), but I think a footnote on the table should indicate the reference to the previous study.

  • While the factors identified by the authors were expected, it does not decrease from the value of the manuscript, the sample size was representative and the statistical analysis was adequate to reach the conclusion, and I believe that it adds some value to the field of public health and HIV care.

Author Response

We have attached point by point response to the reviewers' comments and the cover letter.

Reviewer 2 Report

Co

This manuscript by Tesfay F.H. et.al investigate the relapse of undernutrition and associated factors among people living with HIV in the Tigray region of Ethiopia. The finding of this study suggested that the sociodemographic and cultural characteristics; low education, poor employment, rural residence, involvement in a community support group were predictors of relapse. The manuscript is well presented/organized, and it is a part of Ph.D. thesis of first author as indicated in Acknowledge section.

  1. Most of the introduction parts seem like discussion. The introduction should give the readers the beginning of HIV and nutritional care. Therefore, I would suggest kindly move line no.39-60 to the discussion section and kindly redraft the introduction.
  2. Results should be clearly presented. The numbering of subheading is confusing. Please consider providing a dedicated result section. Please revisit subheading 2.7. to 2.16. kindly change it to main heading of Results section eg. 3.1., 3.2., and so on.
  3. In the result section, unfortunately, although very detailed, this study includes numerous redundancies and repetitions. Some parts just read as a summary of data gathered, without clear structural and connecting lines. Please consider reorganizing the results in a more streamlined way.
  4. I strongly feel that a data comparing the ART regime and Frequency of relapses of undernutrition will be benefitted. Please consider including these and compare these with the pre-ART. These data can also be included in the table format. This will also shed light if ART regime is also associated with relapse of undernutrition.

Author Response

Point by point response to the reviewers' comments and the cover letter are attached in a separate file. 
